# NBF2, an Algal Fiber-Rich Formula, Reverses Diabetic Dyslipidemia and Hyperglycemia In Vivo

**DOI:** 10.3390/ijms251910828

**Published:** 2024-10-09

**Authors:** Nlandu Roger Ngatu, Akram Hossain, Nao Maruo, Steeve Akumwami, Asadur Md. Rahman, Masamitsu Eitoku, Kanae Kanda, Akira Nishiyama, Narufumi Suganuma, Tomohiro Hirao

**Affiliations:** 1Department of Public Health, Kagawa University Faculty of Medicine, Kagawa 761-0793, Japan; kanda.kanae@kagawa-u.ac.jp; 2Department of Medical Pharmacology, Kagawa University Faculty of Medicine, Kagawa 761-0793, Japan; akram.hossain@kagawa-u.ac.jp (A.H.); rahman.md.asadur@kagawa-u.ac.jp (A.M.R.); nishiyama.akira@kagawa-u.ac.jp (A.N.); 3Department of Environmental Medicine, Kochi University School of Medicine, Nankoku 783-8505, Japan; jm-n.maruo@kochi-u.ac.jp (N.M.); meitoku@kochi-u.ac.jp (M.E.); nsuganuma@kochi-u.ac.jp (N.S.); 4Department of Anesthesiology, Kagawa University Faculty of Medicine, Kagawa 761-0793, Japan; s21d703@kagawa-u.ac.jp

**Keywords:** algal formula, dyslipidemia, hyperglycemia, peroxisome proliferation-activated receptor, sirtuin, type 2 diabetes

## Abstract

*Ulva prolifera*, known as Aonori in Japan, is an edible alga species that is mass-cultivated in Japan. Supplementation with Aonori-derived biomaterials has been reported to enhance metabolic health in previous studies. This was an experimental study that evaluated the metabolic health effects of NBF2, a formula made of algal and *junos Tanaka citrus*-derived biomaterials, on obesity and type 2 diabetes (T2DM). We used 18 obese and hyperglycemic Otsuka Long-Evans Tokushima Fatty (OLETF) rats that were assigned randomly to three groups of six animals: a high-dose NBF2 drink (20 mg/kg) group, a low-dose (10 mg/kg) NBF2 drink group and the control group that received 2 mL of tap water daily for a total of six weeks. We also used eight LETO rats as the normal control group. In addition to the glucose tolerance test (OGTT), ELISA and real-time PCR assays were performed. High-dose and lowdose NBF2 improved insulin sensitivity, as well as glycemic and lipid profiles, as compared with control rats. The OGTT showed that both NBF2 groups and LETO rats had normalized glycemia by the 90-min time-point. NBF2 up-regulated PPARα/γ-mRNA and Sirt2-mRNA gene expressions in BAT and improved the blood pressure profile. These findings suggest that the NBF2 formula, which activates PPAR-α/γ mRNA and Sirt2-mRNA, may reverse dyslipidemia and hyperglycemia in T2DM.

## 1. Introduction

Drugs that are agonists of peroxisome proliferation-activated receptors (PPARs) may be effective in the management of lifestyle-related metabolic disorders such as dyslipidemia and type 2 diabetes mellitus (T2DM) [1]. The latter condition is characterized by an inadequate pancreatic beta cell response to insulin resistance. Recently, T2DM management has undergone a major conceptual change, with more emphasis on a patient-centered care approach and the safety profile of antihyperglycemic agents [1,2,3]. In T2DM, the loss of insulin function is reported to be closely related to adiponectin (APN) deficiency. Thus, APN replenishment in T2DM patients might reverse this condition and improve their metabolic health. APN, a protein produced by adipocytes, has been reported to stimulate fatty acid oxidation, improve insulin sensitivity and inhibit neoglucogenesis, resulting in an improvement in energy homeostasis [4]. The insulin-sensitizing effect of APN has been linked to PPAR activation [5,6]. APN also plays a role in tissue regeneration [7,8].

The up-regulation of APN production to correct its deficiency and the restoration of the function of cells involved in energy homeostasis have motivated our research. Natural compounds from *Radix astragali* [9,10] and omega-3 poly-unsaturated fatty acids have been reported to modulate APN production and improve the metabolic health markers of high-risk subjects [11]. We discovered APN-modulating biomaterials (Ngatu Bio Formulas or NBF) containing edible green marine algal species mass-cultivated in Japan, known as Aonori and Aosanori, with beneficial metabolic health effects [12,13].

This study evaluated the effects of NBF2, a formula composed of Aonori alga and *Junos citrus* peel-derived biomaterials, on metabolic health markers in obesity and T2DM model rats.

## 2. Results

### 2.1. Obesity and Diabetic Hyperglycemia Establishment in OLETF Rat Model and Effects of Daily Intake of NBF2 on Body Weight (BW) and Fasting Blood Glucose (FBG)

As shown in Figure 1, treatment with NBF2 significantly reduced weight gain in OLETF rats that received both high and low doses of NBF2 versus vehicle-treated OLETF controls (*p* < 0.05) (Figure 1A). Additionally, both low- and high-dose NBF2-treated OLETF rats had reduced fasting blood glucose (FBG) as compared with OLETF controls (*p* < 0.01), which was normalized within the third week of treatment (Figure 1B).

In this study, an oGTT was performed by administering 0.002g of glucose/1g body weight); FBG was measured at baseline (T0), 15 min (T15), 30 min (T30), 60 min (T60), and 120 min (T120). Moreover, the decrease in FBG was greater in high- (*p* < 0.01) and low-dose (*p* < 0.05) NBF2-treated rat groups, as well as in normal rats (*p* < 0.001; vs. controls) (Figure 1C).

Furthermore, only the normal (LETO) rat group had a significantly lower daily calorie intake (vs. control group; *p* = 0.000). No significant difference was observed when comparing NBF2-treated rats and the controls (Appendix A).

### 2.2. Effects of NBF2 on Plasma Cholesterol, Triglycerides, Adiponectin, Insulin Resistance and Blood Pressure Profile

Figure 2 shows the plasma levels of cholesterols, triglycerides (TGs) and the blood pressure profile. The high-dose NBF2 group had significantly lower total cholesterol (Figure 2A), LDL-c (*p* < 0.05) (Figure 2B) and TG levels (*p* < 0.01) (Figure 2C) compared to the control group. Similarly, the low-dose NBF2 group also showed significantly reduced total cholesterol, TGs and LDL-c (*p* < 0.05). On the other hand, the HDL-c level was significantly higher in the high-dose NBF2 group (vs. control group; *p* < 0.05). Furthermore, significantly reduced blood pressure was observed in the high- and low-dose NBF2 groups for DBP and MBP compared to the control group (*p* < 0.05) (Figure 2E–G).

ELISA assays showed significantly higher APN levels in both NBF2 groups compared to the control group (*p* < 0.05). Regarding plasma insulin, a relatively low level was observed in NBF2-treated rats but this was not significant (Figure 3A,B). Furthermore, a significant reduction in HOMA-IR was observed in the NBF2-treated groups compared to the control group (−17% versus −3% change, respectively; *p* < 0.05) (Figure 3C).

## 3. Messenger RNA (mRNA) Gene Expression Levels of Metabolic Biomarkers

Real-time PCR assays were performed to determine the mRNA gene expression levels of metabolic biomarkers. Table 1 presents the list of primers of rat biomarkers and the corresponding mRNA gene sequences. Leptin and fatty acid synthase (FASN) are linked to the development of obesity and related metabolic disorders. It was observed that rats treated with NBF2 had significantly reduced leptin mRNA expression in brown (BAT) and white adipose tissue (WAT) and liver specimens, as compared with vehicle-treated OLETF controls (*p* < 0.05) (Figure 4A–C). Similarly, NBF2-treated rats had reduced FASN mRNA gene expression in both BAT and WAT specimens (Figure 4D), suggesting an inhibitory effect of NBF2 on fat accumulation.

Unlike FASN and leptin, sirtuin 2 (Sirt2) and the peroxisome proliferator-activated receptors alpha and gamma (PPAR-α/γ) are known to improve obesity-associated metabolic disorders and are considered therapeutic targets for diseases such as T2DM. We observed that treatment with NBF2 caused a marked increase in Sirt2-mRNA expression in the rat BAT (*p* < 0.01) and WAT (*p* < 0.05), as well as an increase in PPAR–γ and PPAR–α mRNA expressions in OLETF rat BAT as compared with the vehicle in the control rats (Figure 4G,H). The lowdose NBF2 group also had reduced APN-mRNA gene expression in BAT and APN-mRNA gene expression in WAT. ATPase sarcoplasmic/endoplasmic reticulum Ca^2+^, also known as ATPase 2a2 or SERCA2, is an enzyme that plays an important metabolic role as a glucose transport regulator; it is down-regulated in the adipocytes of diabetic animals and humans. We found that BAT ATP2a2-mRNA expression was up-regulated in NBF2-treated rats (Figure 4M–O) (vs. control group; *p* < 0.05). Consequently, the dual PPARα/γ and Sirt2 modulatory effects of NBF2 observed in this study suggest an improvement in lipid and glucose metabolism.

In the histopathological examination, white adipose tissue (WAT) and liver sections showed the following results: Large adipocytes were mostly observed in WAT specimens from OLETF control rats (Appendix A) compared to in specimens from normal rats (Appendix A) and NBF2-treated animals (Appendix A). Regarding BAT specimens, the presence of smaller adipocytes mixed with large adipocytes was observed in OLETF controls (Appendix A), but larger ones were predominant in BAT from rats treated with NBF2 (Appendix A) and in LETO rats (Appendix A). Images of blood sinusoidal dilation and hepatic steatosis with lipid droplets were mostly observed in liver specimens from OLETF control rats (Appendix A), whereas these changes were either reduced and inexistent in liver specimens from NBF2-treated OLETF (Appendix A) and normal rats (Appendix A).

## 4. Discussion

In the present experimental study, we explored the metabolic health effects of low and high (normal) doses of NBF2, a formula containing marine Aonori alga and *junos* Tanaka *citrus*-derived biomaterials, on lipid and glucose metabolism in obese and hyperglycemic OLETF rats. NBF2 treatment increased APN production.

Previously, we reported that algal biomaterial (NBF1), which represents 60% of NBF2, induced a 2- to 3-fold increase in APN in human subjects [12]. On the other hand, *junos citrus* peel, another component of NBF2 that represents 40%, is mainly composed of a potent anti-inflammatory compound, limonene [14]. Thus, the formula used in this study can be considered a potent PPAR modulator, an APN modulator and an anti-inflammatory agent that exerts beneficial effects on glucose and lipid metabolism.

In our study, NBF2 intake prevented weight gain in OLETF rats compared to the vehicle. Moreover, NBF2 could improve insulin sensitivity and also normalize blood glucose in OLETF rats to a level similar to that of lean LETO rats across the experiment. The latter finding can be explained by the increased brown fat mass, the relatively low FASN mRNA expression and the enhanced APN production in NBF2-treated OLETF rats. APN is known to ameliorate insulin resistance and glucose intolerance. By restoring insulin sensitivity through the up-regulation of APN production, NBF2 could reverse hyperglycemia. Recently, we have reported that in addition to its anti-inflammatory effect, NBF-containing algal biomaterial has a potential to improve the glycemic profile in individuals with a high cardiometabolic risk and in diabetic patients as well [13].

This study also showed an enhanced lipid metabolism and BP profile in obese and hyperglycemic rats treated with NBF2. The observed improvement in lipid profile, possibly intertwined with Sirt2 activation, may explain the slowing of weight gain and the improved BP. Recently, evidence has been accumulated suggesting that not only PPARs but also sirtuin activators can be utilized as novel therapeutic targets for metabolic disorders [15,16]. Sirt2, known as an NAD+-dependent deacetylase and which was activated by NBF2 in our study, plays an important role in terms of lipid metabolism regulation and improving liver function [17,18,19].

Compared to WAT, BAT has been reported to produce a larger amount of circulating APN. On the other hand, PPARs are known to induce the browning of white fat and regulate brown adipocyte thermogenic function [20,21]. The fiber-rich algal component of NBF2 has displayed a good safety profile, has exerted anti-inflammatory activity in humans in previous studies [22,23] and seems to have influenced PPAR activity in BAT in the present study. This suggests that NBF2 could be a potential natural agent with beneficial metabolic health effects. Further investigations are needed to clarify the relationship between NBF2 and PPARs.

Nonetheless, this study has some limitations. The effects of NBF2 on other metabolic markers such as glucagon-like protein-1 receptor (GLP-1R) have not been explored. Drugs that inhibit GLP-1R have recently emerged as effective therapeutic agents for T2DM, as they stimulate insulin secretion, causing a decrease in blood glucose [24].

Taken together, the findings from this study suggest that the NBF2 formula, which increased BAT, improved glucose tolerance and alleviated insulin resistance in this experiment, has a potential to reverse dyslipidemia and hyperglycemia in T2DM.

## 5. Materials and Methods

### 5.1. Animals and Intervention

The experimental protocol, as well as the animal care procedures, were carried out according to the guidelines for the care and use of animals of the Experimental Animal Center of Kagawa University (ethical approval number: 20656-1), Japan. Six-week-old male Otsuka Long-Evans Tokushima Fatty (OLETF) and genetic control Long-Evans Tokushima Otsuka (LETO) rats were purchased from Japan SLC Inc. (Shizuoka, Japan) and maintained under a controlled temperature (24 ± 2 °C) and humidity (55 ± 5%), with a 12 h light/dark cycle, at the Animal Center of Kagawa University in Japan. The rats were fed with MF chow and had access to tap water ad libitum. They were used when they reached 10 weeks of age. It has been reported that OLETF rats exhibit metabolic syndrome and prediabetes from the age of 10 weeks of age and, later on, the type 2 diabetic phase.

The NBF2-based drink was a gift from Kochi Ice Co., Ltd. (Kochi, Japan), a food processing factory located in Kochi prefecture, Japan. It is a food-based formula containing 60% ulvan-rich alga powder and 40% *Junos* Tanaka *citrus* peel powder. The chemical composition of the two biomaterials that make up the NBF2 formula is as follows:(1)The Aonori (*Ulva prolifera*) algal biomaterial used in this study is rich in fiber (~65%), and its main bioactive component is ulvan [12], followed by protein (9~14%), omega-3 and omega-6 fatty acids (10.4~10.9%), and small amounts of flavonoids, carotenoids and minerals;(2)The *junos citrus* peel biomaterial is rich in limonene (68~70%), other terpenoids such as γ-terpinene (11.4~12.5%), β–phellandrene (4.6~5.4%), myrcene (3.0~3.2%) and α−pinene (2.3~21.7%), and small amounts of aromatic compounds [12,15].

### 5.2. Experimental Protocol

In this experiment, a total of 26 rats were used, including 18 OLETF and 8 LETO (normal) rats. LETO rats were used as normal controls; prediabetic OLETF rats were randomly divided into 3 groups at week 10 of age (baseline) based on body weight and fasting blood glucose (FBG) (Figure 5):(1)Group 1: 2 mL H_2_O (vehicle)-treated OLETF rats or OLETF controls (*n* = 6);(2)Group 2: 20 mg/kg/day NBF2-treated OLETF rats (*n* = 6);(3)Group 3: 10 mg/kg/day NBF2-treated OLETF rats (*n* = 6).

**Figure 5 ijms-25-10828-f005:**
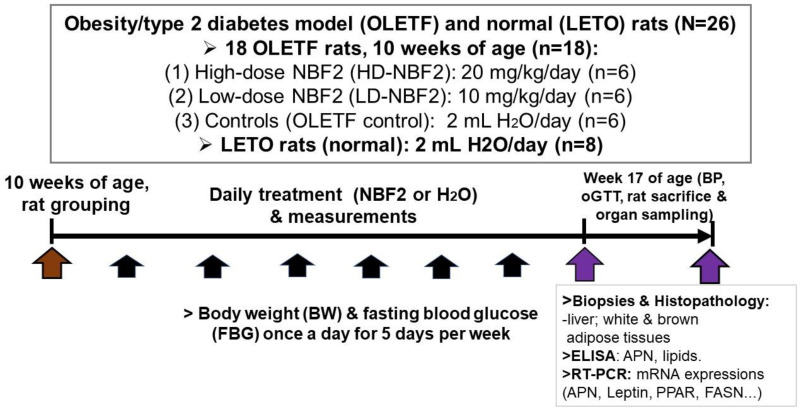
The experimental protocol. Figure 1 shows the procedures used in this study and the measurements performed to obtain the clinical, laboratory (ELISA, RT-PCR assays) and pathological data. **Legend**: APN, adiponectin; PPAR, peroxisome proliferator-activated receptor; FASN, fatty acid synthetase; mRNA, messenger ribonucleic acid; oGTT, glucose tolerance test. Brown arrow shows that start of experiment; purple arrows indicate the week where oGTT and biospecimen sampling were performed for bioassays, black arrows indicate the period in which daily measurements of body weight and blood glucose were carried out.

Daily calorie intake was measured on a weekly basis starting from week 10 of age; the mean value of calorie intake calculated for each group was estimated based on the energy provided by the MF chow and NBF2-containing nutrients. Furthermore, fasting blood glucose (FBG) was measured with the use of a freestyle glucose meter (Glucotest AI, Sanwa Biochemical Research Center, Kanagawa, Japan) from the tip of the rat tail once a week. Body weight was also measured on a weekly basis.

### 5.3. Measurements

The oral glucose tolerance test (OGTT) was carried out in duplicate within the last week of the experiment when the rats were 16 weeks old, as described previously [14]. Briefly, the rats were fasted for 12 h, and 0 min blood glucose was measured. Then, the rats were each fed 2 g/kg of 50% glucose solution by gavage; blood glucose was measured again at 30, 60 and 120 min.

Blood pressure (BP) and heart rate (HR) were measured using an automatic device (BP-98A, Softron, Tokyo, Japan), following the tail-cuff method. Briefly, a photo-elective sensor, which is attached to the animal tail by a piece of rubber tubing, detects the pulse volume oscillations through the cuff pressure and pulse waves. The pulse waves are detected as systolic BP (SBP) and mean BP (MBP); these two values are used to calculate the diastolic BP (DBP) automatically with the use of the following formula: DBP = (3 × MBP − SBP)/2. The homeostatic model assessment of insulin resistance (HOMA-IR) was calculated according to the following formula:HOMA-IR = glycemia (mg/dL) × plasma insulin (UI/mL)/405.

### 5.4. Biospecimen Sampling, Histological Examination and Real-Time PCR

On the day of animal sacrifice, the rats were fasted for 12 h and anesthetized; blood was drawn. Blood lipids were measured using a portable lipid analyzer MLA-1 (Changsha Zealson Biotech Co., Ltd., Changsha, China). Plasma adiponectin and insulin measurements were performed by using the enzyme-linked immunosorbent assay (ELISA; SRL biochemical company, Kobe, Japan). Brown adipose tissue (BAT) was collected from the scapula area, whereas white adipose tissue (WAT) was collected from the mesenteric, retroperitoneal and epididymal spaces. All necessary organs taken from the rats were weighted and stored at −80 °C prior to histological examination. The latter was performed by a pathologist. Visceral adipose tissue and liver specimens were formalin-fixed and paraffin-embedded. They were stained with hematoxylin–eosin (HE) prior to observation of the specimen with an all-in-one fluorescence microscope BZ-X800. (Keyence Corp., Osaka, Japan).

The collected adipose tissues were snap-frozen in liquid nitrogen, then stored at −80 °C until processing for RNA extraction. Real-time reverse transcription PCR (RT-PCR) was performed at the laboratory unit of the department of Environmental Medicine, Kochi University Medical School, Japan. RNA was isolated from WAT, FAT and liver tissue by using the phenol–chloroform extraction method, and cDNA was prepared. Messenger RNA (mRNA) expressions of PPAR−α, PPAR−γ, leptin and APN were analyzed in adipose tissue, whereas those of the three markers of lipid metabolism, namely fatty acid synthase (FAS), sirtuin2 (Sirt2) and ATPase sarcoplasmic/endoplasmic reticulum Ca^2+^ transporting 2 (ATP2a2), were analyzed in liver tissue by RT-PCR using ABI Prism 7000 with Power SYBR Green PCR Master Mix (Invitrogen, Tokyo, Japan) and primer sequences from Eurofins genomics (Japan). Group comparisons for mRNA gene expressions of different biomarkers were performed using the relative quantification of genes.

### 5.5. Ethical Considerations and Statistical Analysis of Collected Data

The study protocol was approved by the Ethical Review Board of the Experimental Animal Center of Kagawa University (approval number: 20656-1). All of the procedures were performed according to internationally adopted guidelines for the care and use of laboratory animals. Mean (SD) values were used to present continuous data. Furthermore, Student’s t test was used when comparing two groups of normally distributed data related to the outcome variables. Analysis of variance (ANOVA) for repeated measures or one-way ANOVA, followed by a post hoc Tukey’s test, was used when appropriate. The significance level was set at a *p*-value less than 0.05. Stata statistical software version 15 was used for all statistical analyses (SataCorp, College Station, TX, USA).

## 6. Conclusions

The present study, which is related to one of the newly discovered alga-based adiponectin-modulating formulas, NBF2, showed that its intake caused Sirt2 and dual PPAR activation, up-regulated APN, and improved glucose and lipid metabolism in a rodent model of obesity and T2DM. These findings suggest that NBF2 may be beneficial in preventing and reversing hyperlipidemia and hyperglycemia in at-risk subjects and T2DM patients.

## Figures and Tables

**Figure 1 ijms-25-10828-f001:**
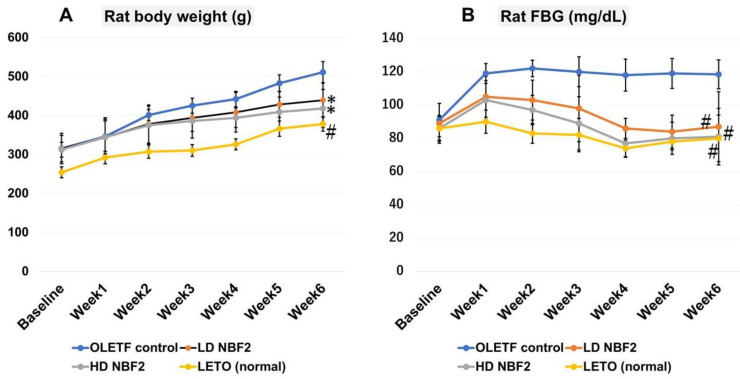
(**A**–**C**): The trend in rat body weight (**A**) and fasting blood glucose across the study period (**B**) and the blood glucose level during the glucose tolerance test (oGTT) (**C**) (*N* = 26). **Legend**: *, *p*-value less than 0.05; #, *p*-value less than 0.01 (by one-way ANOVA); OLETF, Otsuka Long-Evans Tokushima Fatty rats; LETO, Long-Evans Tokushima Otsuka rats.

**Figure 2 ijms-25-10828-f002:**
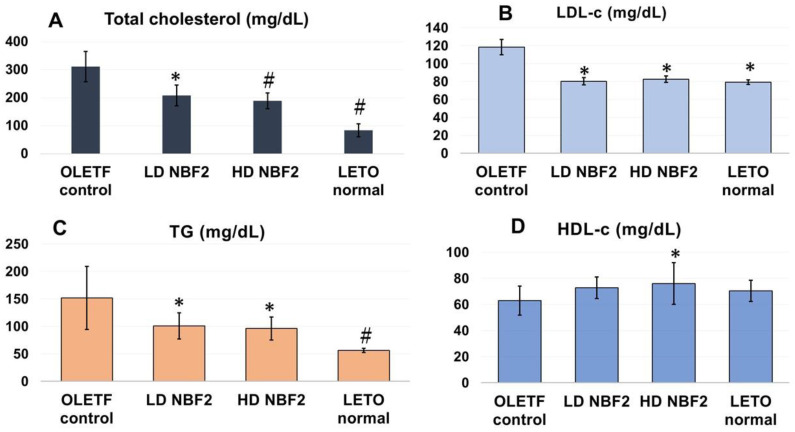
(**A**–**G**) Blood lipid and blood pressure profiles in rats (*N* = 26). **Legend**: *, *p*-value less than 0.05; #, *p*-value less than 0.01 (by one-way ANOVA); OLETF, Otsuka Long-Evans Tokushima Fatty rats; LETO, Long-Evans Tokushima Otsuka rats; T-chol, total cholesterol; LDL-c, low-density lipoprotein cholesterol; HDL-c, high-density lipoprotein cholesterol; TG, triglyceride.

**Figure 3 ijms-25-10828-f003:**
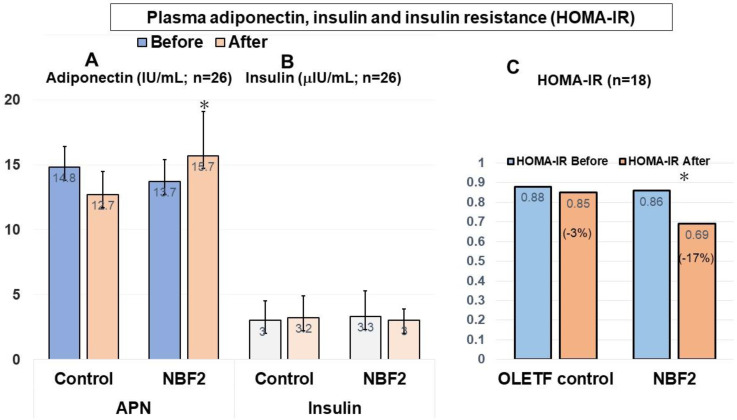
(**A**–**C**) Plasma adiponectin (**A**), insulin (**B**) and HOMA-IR (**C**) (*N* = 24). **Legend**: *, *p* < 0.05; IU, international unit; APN, adiponectin; NBF, Ngatu Bio Formula; HOMA-IR, homeostasis model assessment for insulin resistance.

**Figure 4 ijms-25-10828-f004:**
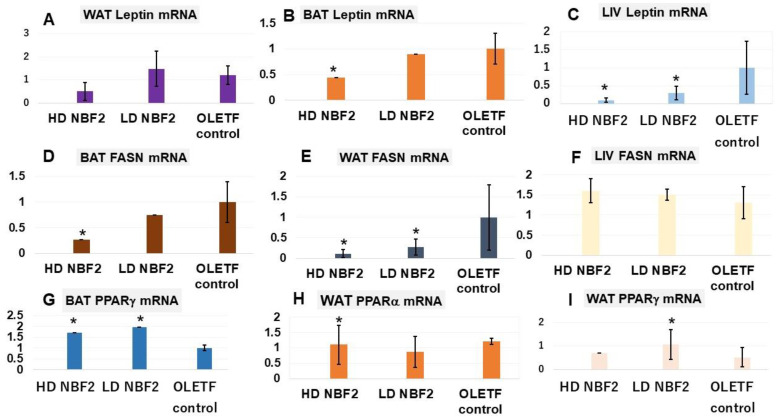
Messenger RNA (mRNA) gene expression of biomarkers in white/brown adipose tissue and liver biopsies (N = 20). **Legend**: *, *p*-value less than 0.05; #, *p*-value less than 0.01 (vs. OLET control; by Student’s *t* test); OLETF, Otsuka Long-Evans Tokushima Fatty rats; LETO, Long-Evans Tokushima Otsuka rats; mRNA, messenger ribonucleic acid; BAT, brown adipose tissue; WAT, white adipose tissue; LIV, liver tissue; APN, adiponectin; FASN, fatty acid synthase; PPAR, peroxisome proliferator-activated receptor; Sirt2; sirtuin 2; ATPa2a or SERCA2, ATPase sarcoplasmic/endoplasmic reticulum Ca^2+^. The unit used is the percentage (%) of the relative mRNA gene expression level for each biomarker.

**Table 1 ijms-25-10828-t001:** List of primers used and corresponding biomarkers’ nucleic acid sequences.

Name	Primer (5′→3′)	Length	GC%	Tm	Target Gene	Amplified Size (bp)	Amplified Area
PPAR-[F] PPAR-[R]	GAGGGCACACGCTAGGAAG GGACTCGAAGTAGCGGACAG	19 20	63 60	60 60	XM_006242152.4	72	GAGGGCACACGCTAGGAAGGGCACACGCGTGCGACTTTCGGGGCCCCTGGAACTGTCCGCTACTTCGA GTCC
PPAR-[F] PPAR-[R]	GCCGCCTCAGATTTGAAAGAA GTCCACAGAGCTGATTCCGA	21 20	48 55	59 60	XM_006237009.4	142	GCCGCCTCAGATTTGAAAGAAGCTGTGAACCACTAATATCCAAGGACATTTTTGAAAACAAGGACTACCCTTTACTGAAATTACCATGGTTGACACAGAGATGCCATTCTGGCCCACCAACTTCGGAATCAGCTCTGTGGAC
APN [F] APN [R]	CAAGCGCTCCTGTTCCTCTT TGGGTCACCCTTAGGACCAA	20 20	55 55	62 62.6	XM_039087986.1	228	CAAGCGCTCCTGTTCCTCTTAATCCTGCCCAGTCATGAAGGGATTACTGCAACCGAAGGGCCAGGAGCTTTGGTCCCTCCACCCAAGGAAACTTGTGCAGGTTGGATGGCAGGCATCCCAGGATATCCTGGTCACAATGGGATACCGGGCCGTGATGGCAGAGATGGCACTCCTGGAGAGAAGGGAGAGAAGGGAGACGCAGGTGTTCTTGGTCCTAAGGGTGACCCA
Sirtuin 2 [F] Sirtuin 2 [R]	GACGAGCTGACCCTTGAAG TCTCCAAGTTTGCATAGAGGC	19 21	58 48	63 63	NM_001399630.1	148	GACGAGCTGACCCTTGAAGGAGTGACACGCTACATGCAGAGCGAGCGCTGTCGCAGGGTCATCTGTTTGGTGGGAGCTGGAATCTCCACATCCGCAGGAATCCCTGACTTCCGCTCCCCATCCACTGGCCTCTATGCAAACTTGGAGA
Atp2a2 [F] Atp2a2 [R]	TTTGGGCAGGATGAGGATG TTGTGGGAAGGTTCAACTCG	19 20	53 50	66 65	NM_001110139.2	128	TTTGGGCAGGATGAGGATGTGACATCAAAGGCTTTTACAGGGCGAGAATTTGATGAATTAGCCCCTCAGCCCAGAGAGACGCCTGCTTAAATGCCCGTTGTTTTGCTCGAGTTGAACCTTCCCACAA
Fasn [F] Fasn [R]		20 20	55 55	64 63	NM_017332.2	214	TCGCTCATGGGTGTGGAAGTGCGCCAGATCCTGGAACGTGAACATGATCTGGTGCTACCCATTCGTGAAGTACGGCAACTCACACTGCGGAAGCTTCAGGAAATGTCCTCCAAGGCTGGCTCAGACACTGAGTTGGCAGCCCCCAAGTCCAAGAATGATACATCCCTGAAGCAGGCCCAGCTGAATCTGAGTATCCTGCTGGTGAACCCTGA
Leptin [F] Leptin [R]	GAGACCCCTGTGTCGGTTC CTGCGTGTGTGAAATGTCATTG	19 22	63 45	60 59	NM_013076.3	139	GAGACCCCTGTGCCGGTTCCTGTGGCTTTGGTCCTATCTGTCCTATGTTCAAGCTGTGCCTATCCACAAAGTCCAGGATGACACCAAAACCCTCATCAAGACCATTGTCACCAGGATCAATGACATTTCACACACGCAG
rGapdh [F] rGapdh [R]	TGATTCTACCCACGGCAAGT AGCATCACCCCATTTGATGT	20 20	50 45	64 65	NM_017008.4	142	TGACTCTACCCACGGCAAGTTCAACGGCACAGTCAAGGCTGAGAATGGGAAGCTGGTCATCAACGGGAAACCCATCACCATCTTCCAGGAGCGAGATCCCGCTAACATCAAATGGGGTGATGCT

**Notes:** APN, adiponectin; F, forward; R, reverse; %, percent.

## Data Availability

The datasets used in this study are available from the corresponding authors.

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
