# Peer review of "NBF2, an Algal Fiber-Rich Formula, Reverses Diabetic Dyslipidemia and Hyperglycemia In Vivo"

_ijms, 2024, doi:10.3390/ijms251910828_

Round 1

Reviewer 1 Report

Comments and Suggestions for Authors

1. Give the full form of APN in the "Introduction" once it is mentioned for the first time.

2. Figures and Table 1 is not very clear.

3. Pathological changes were not properly indicated in the histopathological images.

4. what could be the reason for higher calorie intake in higher dose NBF2 group along with a reduced weight gain. Show the data in the manuscript.

5. Quality of the figures are compromised. Replace with high quality images.

6. 'n' values (number of replicates) for each experiment should be mentioned in the figure legends.

7. There is no units for Fig. 5.

8. Authors are encouraged to edit English language in the manuscript.

9. NBF2 stands for what??

10. How do you administered the drink to rats? Mention these details under methods. 

Comments on the Quality of English Language

Must be improved

Author Response

ANSWERS TO COMMENTS FROM REVIEWER 1

  1. Give the full form of APN in the "Introduction" once it is mentioned for the first time.

- Answer1: Thank you. We provide the full term and then the abbreviation in the revised version of the manuscript.

  1. Figures and Table 1 is not very clear.

- Answer2: We provided high resolution figures for the revised manuscript.

  1. Pathological changes were not properly indicated in the histopathological images.

- Answer3: Thank you. In the manuscript, we describe only histopathological changes observed in hematoxylin-eosin (HE) coloration, mainly in relation to fat accumulation. Other coloration techniques were not used in this experiment.

  1. what could be the reason for higher calorie intake in higher dose NBF2 group along with a reduced weight gain. Show the data in the manuscript.

- Answer4: We previously compared high/low dose NBF2 group to the control group (OLETF control) using one-way ANOVA. Now, we performed repeated measure ANOVA and found significantly low-calorie intake only for normal control rats (LETO) versus OLETF controls. We revised the paragraph related to this result. Supplementary Table 1 is provided in the revised manuscript.

  1. Quality of the figures are compromised. Replace with high quality images.

- Answer5: As mentioned above, we provide high resolution figures for the revised manuscript.

  1. 'n' values (number of replicates) for each experiment should be mentioned in the figure legends.

- Answer6: Thank you for mentioning that. We included n values in figure legends as you suggested.

  1. There is no units for Fig. 5.

- Answer7: Thanks for asking. We added the unit in the legend.

  1. Authors are encouraged to edit English language in the manuscript.

- Answer8: Yes, we consider having the manuscript English edited.

  1. NBF2 stands for what??

- Answer9: we included the explanation in the revised manuscript, in the Introduction section. NBF (NBF1, NBF2) is an abbreviation for Ngatu Bio Formula; it combines the name of the research who made the discovery of metabolic health effects of algal bio material-derived Formulae from ulva species grown in shikoku area, Japan.

  1. How do you administered the drink to rats? Mention these details under methods. 

- Answer9: Thank you for this question. 2cc of NBF2 drink was given daily for 5 days a week; the normal and OLETF control groups received similar amount of tap water. We added that is the revised manuscript.

Reviewer 2 Report

Comments and Suggestions for Authors

Authors need to revise their manuscript further as the current one has major flaws. Critically, Figure 2a-2b is missing; All figures have low resolution. All the citations are misplaced. Authors need to thoroughly correct those errors before this manuscript can be further evaluated. I would suggest rejection and resubmission.

Author Response

ANSWERS TO COMMENTS FROM REVIEWER2:

Authors need to revise their manuscript further as the current one has major flaws. Critically, Figure 2a-2b is missing; All figures have low resolution. All the citations are misplaced. Authors need to thoroughly correct those errors before this manuscript can be further evaluated. I would suggest rejection and resubmission.

Answer:

Thank you so much for raising those concerns:

  • 2a-b and Fig.2c are provided in the revised version of the manuscript. Originally, it was in the online version in Preprints site but was not visible after automatic transfer to IJMS/MDPI site.
  • Regarding low resolution of figures and citations that did not follow the journal’s instruction: this paper was automatically transferred from Preprints website to the journal site following an invitation from the IJMS editorial staff; thus, the manuscript formatting and citations styles were not those of IJMS/MDPI journal.

We revised the paper following the journal’s instructions to the authors and improved the image resolution of figures

Round 2

Reviewer 1 Report

Comments and Suggestions for Authors

Route of administration of the drink should be given in the manuscript

Comments on the Quality of English Language

Grammatical errors are there through out the manuscript

Author Response

To Reviewer1

We thank you for suggesting a moderate English language editing and improving the presentation of manuscript sections. We thoroughly revised all sections and are going to have the manuscript English edited.

Reviewer 2 Report

Comments and Suggestions for Authors

The revised manuscript demonstrates improved quality. However, while the Algal Fiber-Rich NBF2 Formula showed slight to moderate effects in OLETF models, the authors should avoid exaggeration in their descriptions. For instance, the claim in the discussion that the formula “also normalizes blood glucose in OLETF rats to a level similar to that of lean LETO rats across the experiment” is inappropriate and could be misleading.

Additionally, the authors rely solely on mRNA data without providing evidence that NBF2 activates PPARs at the molecular biology level. Therefore, the authors should tone down this finding, and it should not be included in the title. Ideally, molecular activation experiments, such as Western blotting, should be conducted to assess related biomarkers for PPARs activation.

Author Response

To Reviewer2

Thank you for all the suggestions.

We improved the methods section and the conclusions as suggested.

We also modified the title as suggested.

Thank you again.